# Flag Aggregator: Distributed Training under Failures and Augmented Losses using Convex Optimization

## Abstract

Modern ML applications increasingly rely on complex deep learning models and large datasets. There has been an exponential growth in the amount of computation needed to train the largest models. Therefore, to scale computation and data, these models are inevitably trained in a distributed manner in clusters of nodes, and their updates are aggregated before being applied to the model. However, a distributed setup is prone to Byzantine failures of individual nodes, components, and software. With data augmentation added to these settings, there is a critical need for robust and efficient aggregation systems. We define the quality of workers as reconstruction ratios $\in (0, 1]$, and formulate aggregation as a Maximum Likelihood Estimation procedure using Beta densities. We show that the Regularized form of log-likelihood wrt subspace can be approximately solved using iterative least squares solver, and provide convergence guarantees using recent Convex Optimization landscape results. Our empirical findings demonstrate that our approach significantly enhances the robustness of state-of-the-art Byzantine resilient aggregators. We evaluate our method in a distributed setup with a parameter server, and show simultaneous improvements in communication efficiency and accuracy across various tasks.

## 1 Introduction

**How to Design Aggregators?** We consider the problem of designing aggregation functions that can be written as optimization problems of the form,

$$\mathcal{A}(g_1, \ldots, g_p) \in \arg\min_{Y \in C} A_{g_1, \ldots, g_p}(Y), \tag{1}$$

where $\{g_i\}_{i=1}^{p} \subseteq \mathbb{R}^n$ are given estimates of an unknown summary statistic used to compute the *Aggregator* $Y^*$. If we choose $A$ to be a quadratic function that decomposes over $g_i$'s, and $C = \mathbb{R}^n$, then we can see $\mathcal{A}$ is simply the standard mean operator. There is a mature literature of studying such functions for various scientific computing applications [1]. More recently, from the machine learning standpoint there has been a plethora of work [2, 3, 4, 5] on designing provably robust aggregators $\mathcal{A}$ for mean estimation tasks under various technical assumptions on the distribution or moments of $g_i$.

**Distributed ML Use Cases.** Consider training a model with a large dataset such as ImageNet-1K [6] or its augmented version which would require data to be distributed over $p$ workers and uses back propagation. Indeed, in this case, $g_i$'s are typically the gradients computed by individual workers at each iteration. In settings where the training objective is convex, the convergence and generalization properties of distributed optimization can be achieved by defining $\mathcal{A}$ as a weighted combination of gradients facilitated by a simple consensus matrix, even if some $g_i$'s are noisy [7, 8]. In a distributed setup, as long as the model is convex we can simultaneously minimize the total iteration or communication complexity to a significant extent i.e., it is possible to achieve convergence

**Estimate Subspace for Aggregation**

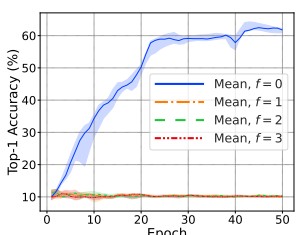

Figure 1: Robust gradient aggregation in our distributed training framework. In our applications, each of the $p$ workers provides gradients computed using a random sample obtained from given training data, derived synthetic data from off-the-shelf Diffusion models, and random noise in each iteration. Our Flag Aggregator (FA) removes high frequency noise components by using few rounds of Singular Value Decomposition of the concatenated Gradient Matrix $G$, and provides new update $Y^*$.

*and* robustness under technical assumptions on the moments of (unknown) distribution from which $g_i$'s are drawn. However, it is still an open problem to determine the optimality of these procedures in terms of either convergence or robustness [9, 10].

**Potential Causes of Noise.** When data is distributed among workers, hardware and software failures in workers [11, 12, 13] can cause them to send incorrect gradients, which can significantly mislead the model [14]. To see this, let's consider a simple experiment with 15 workers, that $f$ of them produce uniformly random gradients. Figure 2 shows that the model accuracy is heavily impacted when $f > 0$ when mean is used to aggregate the gradients.

The failures can occur due to component or software failures and their probability increases with the scale of the system [15, 16, 17]. Reliability theory is used to analyze such failures, see Chapter 9 in [18], but for large-scale training, the distribution of total system failures is not independent over workers, making the total noise in gradients dependent and a key challenge for large-scale training. Moreover, even if there are no issues with the infrastructure, our work is motivated by the prevalence of data augmentation, including hand-chosen augmentations. Since number of parameters $n$ is often greater than number of samples, data augmentation improves the generalization capabilities of large-scale models under technical conditions [19, 20, 21]. In particular, Adversarial training is a common technique that finds samples that are close to training samples but classified as a different class at the current set of parameters, and then use such samples for parameter update purposes [22]. Unfortunately, computing adversarial samples is often difficult [23], done using randomized algorithms [24] and so may introduce dependent (across samples) noise themselves. In other words, using adversarial training paradigm, or the so-called inner optimization can lead to noise in gradients, which can cause or simulate dependent "Byzantine" failures in the distributed context.

Figure 2: Tolerance to $f$ Byzantine workers for a non-robust aggregator (mean).

**Available Computational Solutions.** Most existing open source implementations of $\mathcal{A}$ rely just on (functions of) pairwise distances to filter gradients from workers using suitable neighborhood based thresholding schemes, based on moment conditions [25, 26, 27]. While these may be a good strategy when the noise in samples/gradients is somewhat independent, these methods are suboptimal when the noise is dependent or nonlinear, especially when $n$ is large. Moreover, choosing discrete

hyperparameters such as number of neighbors is impractical in our use cases since they hamper convergence of the overall training procedure. To mitigate the suboptimality of existing aggregation schemes, we explicitly estimate a subspace $Y$ spanned by "most" of the gradient workers, and then use this subspace to estimate that a **sparse** linear combination of $g_i$ gradients, acheiving robustness.

We present a new optimization based formulation for generalized gradient aggregation purposes in the context of distributed training of deep learning architectures, as shown in Figure 1.

**Summary of our Contributions.** From the theoretical perspective, we present a simple Maximum Likelihood Based estimation procedure for aggregation purposes, with novel regularization functions. Algorithmically, we argue that any procedure used to solve Flag Optimization can be directly used to obtain the optimal summary statistic $Y^*$ for our aggregation purposes. **Experimentally**, our results show resilience against Byzantine attacks, encompassing physical failures, while effectively managing the stochasticity arising from data augmentation schemes. In practice, we achieve a *significantly* ($\approx 20\%$) better accuracy on standard datasets. Our **implementation** offers substantial advantages in reducing communication complexity across diverse noise settings through the utilization of our novel aggregation function, making it applicable in numerous scenarios.

## 2 Robust Aggregators as Orthogonality Constrained Optimization

In this section, we first provide the basic intuition of our proposed approach to using subspaces for aggregation purposes using linear algebra, along with connections of our approach standard eigendecomposition based denoising approaches. We then present our overall optimization formulation in two steps, and argue that it can be optimized using existing methods.

### 2.1 Optimal Subspace Hypothesis for Distributed Descent

We will use lowercase letters $y, g$ to denote vectors, and uppercase letters $Y, G$ to denote matrices. We will use **boldfont 1** to denote the vector of all ones in appropriate dimensions. Let $g_i \in \mathbb{R}^n$ is the gradient vector from worker $i$, and $Y \in \mathbb{R}^{n \times m}$ is an orthogonal matrix representation of a subspace that gradients could live in such that $m \leq p$. Now, we may interpret each column of $Y$ as a basis function that act on $g_i \in \mathbb{R}^n$, i.e., $j-$th coordinate of $(Y^T g)_j$ for $1 \leq j \leq m$ is the application of $j-$th basis or column of $Y$ on $g$. Recall that by definition of dot product, we have that if $Y_{:,j} \perp x$, then $(Y^T g)_j$ will be close to zero. Equivalently, if $g \in \text{span}(Y)$, then $(Y^T g)^T Y^T g$ will be bounded away from zero, see Chapter 2 in [28]. Assuming that $G \in \mathbb{R}^{n \times p}$ is the gradient matrix of $p$ workers, $YY^T G \in \mathbb{R}^{n \times p}$ is the reconstruction of $G$ using $Y$ as basis. That is, $i^{th}$ column of $Y^T G$ specifies the amount of gradient from worker $i$ as a function of $Y$, and high $l_2$ norm of $Y^T g_i$ implies that there is a basis in $Y$ such that $Y \not\perp g_i$. So it is easy to see that the average over columns of $YY^T G$ would give the final gradient for update.

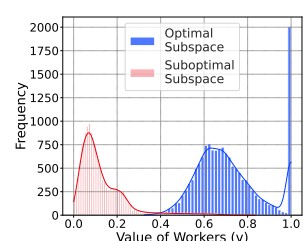

Figure 3: Distributions of Explained Variances on Minibatches

**Explained Variance of worker $i$.** If we denote $z_i = Y^T g_i \in \mathbb{R}^m$ representing the transformation of gradient $g_i$ to $z_i$ using $Y$, then, $0 \leq \|z_i\|_2^2 = z_i^T z_i = (Y^T g)^T Y^T g = g_i^T YY^T g_i$ is a scalar, and so is equal to its trace $\text{tr}\left(g_i^T YY^T g_i\right)$. Moreover, when $Y$ is orthogonal, we have $0 \leq \|z_i\|_2 = \|Y^T g_i\|_2 \leq \|Y\|_2 \|g_i\|_2 \leq \|g_i\|_2$ since the operator norm (or largest singular value) $\|Y\|_2$ of $Y$ is at most 1. Our main idea is to use $\|z_i\|_2^2, \|g_i\|_2^2$ to define the quality of the subspace $Y$ for aggregation, as is done in some previous works for Robust Principal Component Estimation [29] – the quantity $\|z_i\|_2^2 / \|g_i\|_2^2$ is called as *Explained/Expressed* variance of subspace $Y$ wrt $i-$th worker [30, 31] – we refer to $\|z_i\|_2^2 / \|g_i\|_2^2$ as the "value" of $i-$th worker. In Figure 3, we can see from the spike near 1.0 that if we choose the subspace carefully (blue) as opposed to merely choosing the mean gradient (with unit norm) of all workers, then we can increase the value of workers.

**Advantages of Subspace based Aggregation.** We can see that using subspace $Y$, we can easily: 1. handle different number of gradients from each worker, 2. compute gradient reconstruction $YY^T G$ efficiently whenever $Y$ is constrained to be orthogonal $Y = \sum_i y_i y_i^T$ where $y_i$ is the $i-$th column of $Y$, otherwise have to use eigendecomposition of $Y$ to measure explained variance which can be time consuming. In (practical) distributed settings, the quality (or noise level) of gradients in

each worker may be different, **and/or** each worker may use a different batch size. In such cases, handcrafted aggregation schemes may be difficult to maintain, and fine-tune. For these purposes with an Orthogonal Subspace $Y$, we can simply reweigh gradients of worker $i$ according to its noise level, **and/or** use $g_i \in \mathbb{R}^{n \times b_i}$ where $b_i$ is the batch size of $i-$th worker with $\text{tr}(z_i^T z_i)$ instead.

**Why is optimizing over subspaces called "Flag" Optimization?** Recent optimization results suggest that we can exploit the finer structure available in Flag Manifold to specify $Y$ more precisely [32]. For example, $Y \in \mathbb{R}^{m \times n}$ can be parametrized directly as a subspace of dimension $m$ or as a nested sequence of $Y_k \in \mathbb{R}^{m_k \times n}, k = 1, ..., K$ where $m_k < m_{k+1} \leq p \leq n$ such that $\text{span}(Y_k) \subseteq \text{span}(Y_{k+1})$ with $Y_K \in \mathbb{R}^{m \times n}$. When $m_{k+1} = m_k = 1$, we have the usual (real) Grassmanian Manifold (quotient of orthogonal group) whose coordinates can be used for optimization, please see Section 5 in [33] for details. In fact, [34] used this idea to extend median in one-dimensional vector spaces to different finite dimensional *subspaces* using the so-called chordal distance between them. In our distributed training context, we use the explained variance of each worker instead. Here, workers may specify dimensions along which gradient information is relevant for faster convergence – an advantage currently not available in existing aggregation implementations – which may be used for smart initialization also. *We use "Flag" to emphasize this additional nested structure available in our formulation for distributed training purposes.*

## 2.2 Approximate Maximum Likelihood Estimation of Optimal Subspace

Now that we can evaluate a subspace $Y$ on individual gradients $g_i$, we now show that finding subspace $Y$ can be formulated using standard maximum likelihood estimation principles [35]. Our formulation reveals that regularization is critical for aggregation especially in distributed training. In order to write down the objective function for finding optimal $Y$, we proceed in the following two steps:

**Step 1.** Assume that each worker provides a single gradient for simplicity. Now, denoting the value of information $v$ of worker $i$ by $v_i = \frac{z_i^T z_i}{g_i^T g_i}$, we have $v_i \in [0, 1]$. Now by assuming that $v_i$'s are observed from Beta distribution with $\alpha = 1$ and $\beta = \frac{1}{2}$ (for simplicity), we can see that the likelihood $\mathbb{P}(v_i)$ is,

$$\mathbb{P}(v_i) := \frac{(1 - v_i)^{-\frac{1}{2}}}{B(1, \frac{1}{2})} = \frac{\left(1 - \frac{z_i^T z_i}{g_i^T g_i}\right)^{-\frac{1}{2}}}{B(1, \frac{1}{2})}, \tag{2}$$

where $B(a, b)$ is the normalization constant. Then, the total log-likelihood of observing gradients $g_i$ as a function of $Y$ (or $v_i$'s) is given by taking the log of product of $\mathbb{P}(v_i)$'s as (ignoring constants),

$$\log \left(\prod_{i=1}^{p} \mathbb{P}(v_i)\right) = \sum_{i=1}^{p} \log \left(\mathbb{P}(v_i)\right) = -\frac{1}{2} \sum_{i=1}^{p} \log(1 - v_i). \tag{3}$$

**Step 2.** Now we use Taylor's series with constant $a > 0$ to approximate individual worker log-likelihoods $\log(1 - v_i) \approx a(1 - v_i)^{\frac{1}{a}} - a$ as follows: first, we know that $\exp\left(\frac{\log(v_i)}{a}\right) = v_i^{\frac{1}{a}}$. On the other hand, using Taylor expansion of $\exp$ about the origin (so large $a > 1$ is better), we have that $\exp\left(\frac{\log(v_i)}{a}\right) \approx 1 + \frac{\log(v_i)}{a}$. Whence, we have that $1 + \frac{\log(v_i)}{a} \approx v_i^{\frac{1}{a}}$ which immediately implies that $\log(v_i) \approx a v_i^{\frac{1}{a}} - a$. So, by substituting the Taylor series approximation of $\log$ in Equation 3, we obtain the *negative* log-likelihood approximation to be *minimized* for robust aggregation purposes as,

$$-\log \left(\prod_{i=1}^{p} \mathbb{P}(v_i)\right) \approx \frac{1}{2} \sum_{i=1}^{p} \left(a\left(1 - v_i\right)^{\frac{1}{a}} - a\right), \tag{4}$$

where $a > 1$ is a sufficiently large constant. In the above mentioned steps, the first step is standard. Our key insight is using Taylor expansion in (4) with a sufficiently large $a$ to eliminate $\log$ optimization which are known to be computationally expensive to solve, and instead solve *smooth $\ell_a, a > 1$ norm* based optimization problems which can be done efficiently by modifying existing procedures [36].

**Extension to general beta distributions, and gradients** $\alpha > 0, \beta > 0, g_i \in \mathbb{R}^{n \times k}$**.** Note that our derivation in the above two steps can be extended to any beta shape parameters $\alpha > 0, \beta > 0$ – there will be two terms in the final negative log-likelihood expression in our formulation (4), one for each $\alpha, \beta$. Similarly, by simply using $v_i = \text{tr}\left(g_i^T Y Y^T g_i\right)$ to define value of worker $i$ in equation (2), and then in our estimator in (4), we can easily handle multiple $k$ gradients from a single worker $i$ for $Y$.

**Algorithm 1** Distributed SGD with proposed Flag Aggregator (FA) at the Parameter Server

---

**Input:** Number of workers $p$, loss functions $l_1, l_2, ..., l_p$, per-worker minibatch size $B$, learning rate schedule $\alpha_t$, initial parameters $w_0$, number of iterations T

**Output:** Updated parameters $w_T$ from any worker

1 **for** $t = 1$ *to* $T$ **do**
2     **for** $\mathfrak{p} = 1$ *to* $p$ *in parallel on machine* $\mathfrak{p}$ **do**
3         **Select a minibatch:** $i_{\mathfrak{p},1,t}, i_{\mathfrak{p},2,t}, ..., i_{\mathfrak{p},B,t}$    $g_{\mathfrak{p},t} \leftarrow \frac{1}{B} \sum_{b=1}^{B} \nabla l_{i_{\mathfrak{p},b,t}}(w_{t-1})$
4     $G_t \leftarrow \{g_{1,t}, \cdots, g_{p,t}\}$ `// Parameter Server receives gradients from` $p$ `workers`
5     $\hat{Y}_t \leftarrow \text{IRLS}(\hat{G}_t)$ with $\hat{G}_t = G_t + \lambda \nabla \mathcal{R}(Y)\mathbf{1}^T$ `// Do IRLS at the Parameter Server for` $\hat{Y}$
6     **Obtain gradient direction** $d_t$**:** $d_t = \frac{1}{p}\hat{Y}_t\hat{Y}_t^T G_t \mathbf{1}$ `// Compute, Send` $d_t$ `to all` $p$ `machines`
7     **for** $\mathfrak{p} = 1$ *to* $p$ *in parallel on machine* $\mathfrak{p}$ **do**
8         **update model:** $w_t \leftarrow w_{t-1} - \alpha_t \cdot d_t$

9 **Return** $w_T$

---

### 2.3 Flag Aggregator for Distributed Optimization

It is now easy to see that by choosing $a = 2$, in equation (4), we obtain the negative loglikelihood (ignoring constants) as $(\sum_{i=1}^{p} \sqrt{1 - g_i^T Y Y^T g_i})$ showing that Flag Median can indeed be seen as an Maximum Likelihood Estimator (MLE). In particular, Flag Median can be seen as an MLE of Beta Distribution with parameters $\alpha = 1$ and $\beta = \frac{1}{2}$. Recent results suggest that in many cases, MLE is ill-posed, and regularization is necessary, even when the likelihood distribution is Gaussian [37]. So, based on the Flag Median estimator for subspaces, we propose an optimization based subspace estimator $Y^*$ for aggregation purposes. We formulate our Flag Aggregator (FA) objective function with respect to $Y$ as a *regularized* sum of likelihood based (or data) terms in (4) using trace operators $\text{tr}(\cdot)$ as the solution to the following constrained optimization problem:

$$\min_{Y:Y^TY=I} A(Y) := \sum_{i=1}^{p} \sqrt{\left(1 - \frac{\text{tr}\left(Y^T g_i g_i^T Y\right)}{\|g_i\|_2^2}\right)} + \lambda \mathcal{R}(Y) \tag{5}$$

where $\lambda > 0$ is a regularization hyperparameter. In our analysis, and implementation, we provide support for two possible choices for $\mathcal{R}(Y)$:

(1) **Mathematical norms:** $\mathcal{R}(Y)$ can be a form of norm-based regularization other than $\|Y\|_{\text{Fro}}^2$ since it is constant over the feasible set in (5). For example, it could be convex norm with efficient subgradient oracle such as, i.e. element-wise: $\sum_{i=1}^{n} \sum_{j=1}^{m} \|Y_{ij}\|_1$ or $\sum_{i=1}^{m} \|Y_{i,i}\|_1$,

(2) **Data-dependent norms:** Following our subspace construction in Section 2.1, we may choose $\mathcal{R}(Y) = \frac{1}{p-1} \sum_{i,j=1,i\neq j}^{p} \sqrt{\left(1 - \frac{\text{tr}(Y^T (g_i - g_j)(g_i - g_j)^T Y)}{D_{ij}^2}\right)}$ where $D_{ij}^2 = \|g_i - g_j\|_2^2$ denotes the distance between gradient vectors $g_i, g_j$ from workers $i, j$. Intuitively, the pairwise terms in our loss function (5) favors subspace $Y$ that also reconstructs the pairwise vectors $g_i - g_j$ that are close to each other. So, by setting $\lambda = \Theta(p)$, that is, the pairwise terms dominate the objective function in (5). Hence, $\lambda$ regularizes optimal solutions $Y^*$ of (5) to contain $g_i$'s with low pairwise distance in its span – similar in spirit to AggregaThor in [38].

**Convergence of Flag Aggregator (FA) Algorithm 1.** With these, we can state our main algorithmic result showing that our FA (5) can be solved efficiently using standard convex optimization proof techniques. In particular, in supplement, we present a smooth Semi-Definite Programming (SDP) relaxation of FA in equation (5) using the Flag structure. This allows us to view the IRLS procedure in 1 as solving the low rank parametrization of the smooth SDP relaxation, thus guaranteeing fast convergence to second order optimal (local) solutions. Importantly, our SDP based proof works for any degree of approximation of the constant $a$ in equation (4) and only relies on smoothness of the loss function wrt $Y$, although speed of convergence is reduced for higher values of $a \neq 2$, see [39]. We leave determining the exact dependence of $a$ on rate of convergence for future work.

**How is FA aggregator different from (Bulyan and Multi-Krum)?** Bulyan is a strong Byzantine resilient gradient aggregation rule for $p \geq 4f + 3$ where $p$ is the total number of workers and $f$ is

the number of Byzantine workers. Bulyan is a two-stage algorithm. In the first stage, a gradient aggregation rule $R$ like coordinate-wise median [40] or Krum [9] is recursively used to select $\theta = p - 2f$ gradients. The process uses $R$ to select gradient vector $g_i$ which is closest to $R$'s output (e.g. for Krum, this would be the gradient with the top score, and hence the exact output of $R$). The chosen gradient is removed from the received set and added to the selection set $S$ repeatedly until $|S| = \theta$. The second stage produces the resulting gradient. If $\beta = \theta - 2f$, each coordinate would be the average of $\beta$-nearest to the median coordinate of the $\theta$ gradients in $S$. In matrix terms, if we consider $S \in \mathbb{R}^{p \times m}$ as a matrix with each column having one non-zero entry summing to 1, Bulyan would return $\frac{1}{m}\text{ReLU}(GS)\mathbf{1}_m$, where $\mathbf{1}_m \in \mathbb{R}^m$ is the vector of all ones, while FA would return $\frac{1}{p}YY^TG\mathbf{1}_p$. Importantly, the gradient matrix is being right-multiplied in Bulyan, but left-multiplied in FA, before getting averaged. While this may seem like a discrepancy, in supplement we show that by observing the optimality conditions of (5) wrt $Y$, we show that $\frac{1}{m}YY^TG$ can be seen as a right multiplication by a matrix parametrized by lagrangian multipliers associated with the orthogonality constraints in (5). This means it should be possible to combine both approaches for faster aggregation.

# 3  Experiments

In this section, we conduct experiments to test our proposed FA in the context of distributed training in two testbeds. First, to test the performance of our FA scheme solved using IRLS (Flag Mean) on standard Byzantine benchmarks. Then, to evaluate the ability of existing state-of-the-art gradient aggregators we augment data via two techniques that can be implemented with Sci-kit package.

**Implementation Details.**    We implement FA in Pytorch [41], which is popular but does not support Byzantine resilience natively. We adopt the parameter server architecture and employ Pytorch's distributed RPC framework with TensorPipe backend for machine-to-machine communication. We extend Garfield's Pytorch library [42] with FA and limit our IRLS convergence criteria to a small error, $10^{-10}$, or 5 iterations of flag mean for SVD calculation. We set $m = \lceil \frac{p+1}{2} \rceil$.

## 3.1  Setup

**Baselines:** We compare FA to several existing aggregation rules: (1) coordinate-wise **Trimmed Mean** [40] (2) coordinate-wise **Median** [40] (3) mean-around-median **(MeaMed)** [43] (4) **Phocas** [44] (5) **Multi-Krum** [9] (6) **Bulyan** [45].

**Accuracy:** The fraction of correct predictions among all predictions, using the test dataset (top-1 cross-accuracy).

**Testbed:** We used 4 servers as our experimental platform. Each server has 2 Intel(R) Xeon(R) Gold 6240 18-core CPU @ 2.60GHz with Hyper-Threading and 384GB of RAM. Servers have a Tesla V100 PCIe 32GB GPU and employ a Mellanox ConnectX-5 100Gbps NIC to connect to a switch. We use one of the servers as the parameter server and instantiate 15 workers on other servers, each hosting 5 worker nodes, unless specified differently in specific experiments. For the experiments designed to show scalability, we instantiate 60 workers.

**Dataset and model:** We focus on the image classification task since it is a widely used task for benchmarking in distributed training [46]. We train ResNet-18 [47] on CIFAR-10 [48] which has 60,000 $32 \times 32$ color images in 10 classes. For the scalability experiment, we train a CNN with two convolutional layers followed by two fully connected layers on MNIST [49] which has 70,000 $28 \times 28$ grayscale images in 10 classes. We also run another set of experiments on Tiny ImageNet [50] in the supplement. We use SGD as the optimizer, and cross-entropy to measure loss. The batch size for each worker is 128 unless otherwise stated. Also, we use a learning decay strategy where we decrease the learning rate by a factor of 0.2 every 10 epochs.

**Threat models:** We evaluate FA under two classes of Byzantine workers. They can send uniformly random gradients that are representative of errors in the physical setting, or use non-linear augmented data described as below.

**Evaluating resilience against nonlinear data augmentation:** In order to induce Byzantine behavior in our workers we utilize ODE solvers to approximately solve 2 non-linear processes, Lotka Volterra

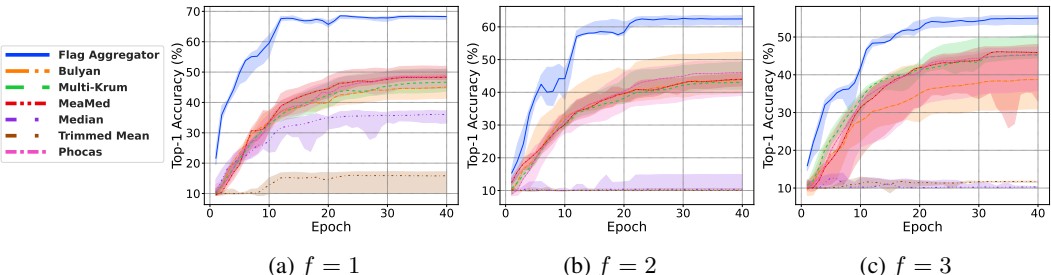

$$(a) \; f = 1 \qquad\qquad (b) \; f = 2 \qquad\qquad (c) \; f = 3$$

Figure 4: Tolerance to the number of Byzantine workers for robust aggregators for batch size 128.

[51] and Arnold's Cat Map [52], as augmentation methods. Since the augmented samples are deterministic, albeit nonlinear functions of training samples, the "noise" is dependent across samples.

In **Lotka Volterra**, we use the following linear gradient transformation of 2D pixels:

$$(x, y) \rightarrow (\alpha x - \beta xy, \delta xy - \gamma y),$$

where $\alpha, \beta, \gamma$ and $\delta$ are hyperparameters. We choose them to be $\frac{2}{3}, \frac{4}{3}, -1$ and $-1$ respectively.

Second, we use a *nonsmooth* transformation called **Arnold's Cat Map** as a data augmentation scheme. Once again, the map can be specified using a two-dimensional matrix as,

$$(x, y) \rightarrow \left( \frac{2x + y}{N}, \frac{x + y}{N} \right) \mod 1,$$

where mod represents the modulus operation, $x$ and $y$ are the coordinates or pixels of images and $N$ is the height/width of images (assumed to be square). We also used a smooth approximation of the Cat Map obtained by approximating the mod function as,

$$(x, y) \rightarrow \frac{1}{n} \left( \frac{2x + y}{(1 + \exp(-m \log(\alpha_1))}, \frac{x + y}{(1 + \exp(-m \log(\alpha_2))} \right),$$

where $\alpha_1 = \frac{2x+y}{n}, \alpha_2 = \frac{x+y}{n}$, and $m$ is the degree of approximation, which we choose to be $0.95$ in our data augmentation experiments.

**How to perform nonlinear data augmentation?** In all three cases, we used SciPy's [53] `solve_ivp` method to solve the differential equations, by using the `LSODA` solver. In addition to the setup described above, we also added a varying level of Gaussian noise to each of the training images. All the images in the training set are randomly chosen to be augmented with varying noise levels of the above mentioned augmentation schemes. We have provided the code that implements all our data augmentation schemes in the supplement zipped folder.

## 3.2 Results

**Tolerance to the number of Byzantine workers:** In this experiment, we show the effect of Byzantine behavior on the convergence of different gradient aggregation rules in comparison to FA. Byzantine workers send random gradients and we vary the number of them from 1 to 3. Figure 4 shows that for some rules, i.e. Trimmed Mean, the presence of even a single Byzantine worker has a catastrophic impact. For other rules, as the number of Byzantine workers increases, filtering out the outliers becomes more challenging because the amount of noise increases. Regardless, FA remains more robust compared to other approaches.

**Marginal utility of larger batch sizes under a fixed noise level:**

We empirically verified the batch size required to identify our optimal $Y^*$ - the FA matrix at each iteration. In particular, we fixed the noise level to $f = 3$ Byzantine workers and varied batch sizes. We show the results in Figure 5. **Our results indicate that, in cases where a larger batch size is a training requirement, FA achieves a significantly better accuracy compared to the existing state of the art aggregators.** This may be useful in some large scale vision applications, see [54, 55] for more details. Empirically, we can already see that our spectral relaxation to identify gradient subspace is effective in practice in all our experiments.

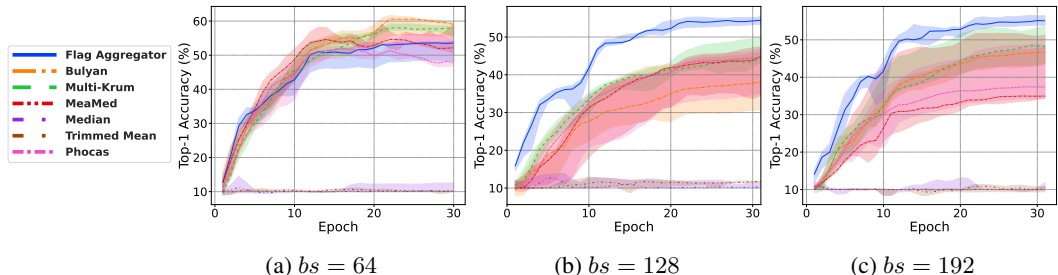

(a) $bs = 64$        (b) $bs = 128$        (c) $bs = 192$

Figure 5: Marginal utility of larger batch sizes under a fixed noise level $f = 3$.

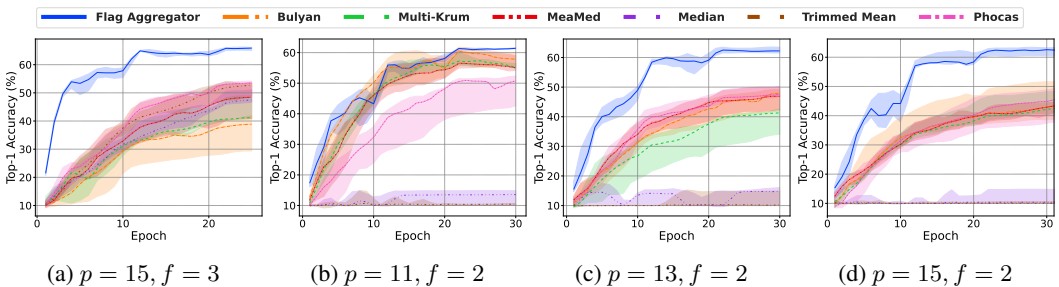

(a) $p = 15, f = 3$    (b) $p = 11, f = 2$    (c) $p = 13, f = 2$    (d) $p = 15, f = 2$

Figure 6: We present results under two different gradient attacks. The attack in (a) corresponds to simply dropping $10\%$ of gradients from $f$ workers. The attacks in (b)-(d) correspond to generic $f$ workers sending random gradient vectors, i.e. we simply fix noise level while adding more workers.

**Tolerance to communication loss:** To analyze the effect of unreliable communication channels between the workers and the parameter server on convergence, we design an experiment where the physical link between some of the workers and the parameter server randomly drops a percentage of packets. Here, we set the loss rate of three links to $10\%$ i.e., there are 3 Byzantine workers in our setting. The loss is introduced using the *netem* queuing discipline in Linux designed to emulate the properties of wide area networks [56]. The two main takeaways in Figure 6a are:

> 1. FA converges to a significantly higher accuracy than other aggregators, and thus is more robust to unreliable underlying network transports.
>
> 2. Considering time-to-accuracy for comparison, FA reaches a similar accuracy in less total number of training iterations, and thus is more robust to slow underlying network transports.

**Analyzing the marginal utility of additional workers.** To see the effect of adding more workers to a fixed number of Byzantine workers, we ran experiments where we fixed $f$, and increased $p$. Our experimental results shown in Figures 6b-6d indicate that our FA algorithm possesses strong resilience property for reasonable choices of $p$.

**The effect of having augmented data during training in Byzantine workers:** Figure 7 shows FA can handle nonlinear data augmentation in a much more stable fashion. Please see supplement for details on the level of noise, and exact solver settings that were used to obtain augmented images.

**The effect of the regularization parameter in FA:** The data-dependent regularization parameter $\lambda$ in FA provides flexibility in the loss function to cover aggregators that benefit from pairwise distances such as Bulyan and Multi-Krum. To verify whether varying $\lambda$ can interpolate Bulyan and Multi-Krum, we change $\lambda$ in Figure 8. We can see when FA improves or performs similarly for a range of $\lambda$. Here, we set $p$ and $f$ to satisfy the strong Byzantine resilience condition of Bulyan, i.e, $p \geq 4f + 3$.

**Scaling out to real-world situations with more workers:** In distributed ML, $p$ and $f$ are usually large. To test high-dimensional settings commonly dealt in Semantic Vision with our FA, we used ResNet-18. Now, to specifically test the scalability of FA, we fully utilized our available GPU servers and set up to $p = 60$ workers (up to $f = 14$ Byzantine) with the MNIST dataset and a simple CNN with two convolutional layers followed by two fully connected layers (useful for simple detection). Figure 9 shows evidence that FA is feasible for larger setups.

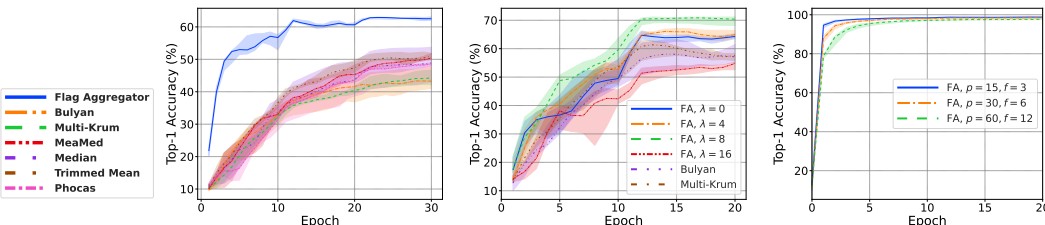

Figure 7: Accuracy of using augmented data in $f = 3$ workers

Figure 8: CIFAR10 with ResNet-18, $p = 7$, and $f = 1$

Figure 9: Scaling FA to larger setups

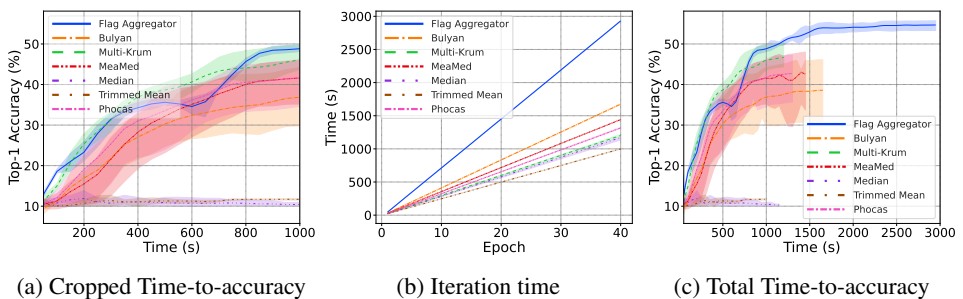

(a) Cropped Time-to-accuracy     (b) Iteration time     (c) Total Time-to-accuracy

Figure 10: Wall clock time comparison

## 4   Discussion and Limitation

**Is it possible to fully "offload" FA computation to switches?** Recent work propose that aggregation be performed entirely on network infrastructure to alleviate any communication bottleneck that may arise [57, 58]. However, to the best of our knowledge, switches that are in use today only allow limited computation to be performed on gradient $g_i$ as packets whenever they are transmitted [59, 60]. That is, *programmability* is restrictive at the moment— switches used in practice have no floating point, or loop support, and are severely memory/state constrained. Fortunately, solutions seem near. For instance, [61] have already introduced support for floating point arithmetic in programmable switches. We may use quantization approaches for SVD calculation with some accuracy loss [62] to approximate floating point arithmetic. Offloading FA to switches has great potential in improving its computational complexity because the switch would perform as a high-throughput streaming parameter server to synchronize gradients over the network. Considering that FA's accuracy currently outperforms its competition in several experiments, an offloaded FA can reach their accuracy even faster or it could reach a higher accuracy in the same amount of time.

**Potential Limitation.** Because in every iteration of FA, we perform SVD, the complexity of the algorithm would be $O(nN_\delta(\sum_{i=1}^{p} k_i)^2)$ with $N_\delta$ being the number of iterations for the algorithm. Figure 10 show the wall clock time it takes for FA to reach a certain accuracy (10a) or epoch(10b) compared to other methods under a fixed amount of random noise $f = 3$ with $p = 15$ workers. Although the iteration complexity of FA is higher, here each iteration has a higher utility as reflected in the time-to-accuracy measures. This makes FA comparable to others in a shorter time span, however, if there is more wall clock time to spare, FA converges to a better state as shown in Figure 10c where we let the same number of total iterations finish for all methods.

## 5   Conclusion

In this paper we proposed Flag Aggregator (FA) that can be used for robust aggregation of gradients in distributed training. FA is an optimization-based subspace estimator that formulates aggregation as a Maximum Likelihood Estimation procedure using Beta densities. We perform extensive evaluations of FA and show it can be effectively used in providing Byzantine resilience for gradient aggregation. Using techniques from convex optimization, we theoretically analyze FA and with tractable relaxations show its amenability to be solved by off-the-shelf solvers or first-order reweighing methods.

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
