# OpenReview forum: "Flag Aggregator: Scalable Distributed Training under Failures and Augmented Losses using Convex Optimization"
_NeurIPS.cc/2023/Conference — Submitted to NeurIPS 2023_

### Official Review · Reviewer_AykZ · 2023-07-05

**Soundness:** 2 fair
**Presentation:** 3 good
**Contribution:** 3 good
**Rating:** 5
**Confidence:** 1

**Summary:**

This work proposes Flag Aggregator (FA) for a more robust aggregation of gradient in data-parallel training. FA formulates gradient aggregation as a Maximum Likelihood Estimation procedure using Beta densities. Theoretically, FA is analyzed using techniques from convex optimization. Empirically, FA demonstrates decent performance against Byzantine failure for image classification tasks (esp. ResNet-18 on CIFAR10) on a 4-GPU cluster networked with 100GbE.

**Strengths:**

+. Proposed a simple Maximum Likelihood Based estimation procedure for aggregation purposes, with novel regularization functions

+. Provided code for reproducibility

+. Well-written: easy to follow

**Weaknesses:**

-. Marginal wall-clock time improvement, maybe due to heavy SVD overhead: e.g., Figure 10

-. Missing benchmark:
1. only two small models are evaluated (e.g., ResNet18 and 2-layer CNN), how about more models like RNNs and larger models like GPT2?

2. only image classification tasks are evaluated (e.g., CIFAR10 and MNIST), not even CIFAR100 nor full ImageNet, how about more tasks like language modeling?

-. Missing modern cluster: 4-GPU cluster with one GPU per machine is not a modern setup for evaluating scalability of distributed training



**Questions:**

*. What if the Byzantine workers send more than just "uniformly random gradients"; how will the FA perform?


**Limitations:**

Yes.

---

> ### Author Rebuttal · Authors · 2023-08-09
>
> **Q.** How about more models like RNNs and larger models like GPT2? And how about more tasks like language modeling?
>
> **Ans.** For Tiny Imagenet experiments in the supplement, we used ResNet-50 as a larger model. In order to store these larger models in the current (or larger) scale of our distributed setup of the experiments, we would require more GPUs that unfortunately we did not have access to. However, our contributions such as presenting a simple Maximum Likelihood Based estimation procedure for aggregation, and significantly better experimental results compared to several baselines still hold in various settings. The extent to which the benefits carry over to larger settings is still open. We hope that our contributions are well received in the research community so that it would open a door for larger (possibly industrial) scale evaluation.
>
> **Q.** What if the Byzantine workers send more than just "uniformly random gradients"; how will the FA perform?
>
> **Ans.** In the main paper, we do have experiments with synthetic data (nonlinear data augmentation routines) and tolerance to communication loss where a percentage of gradients are dropped and zero-ed out at the parameter server. In addition, we have included a figure in the attached PDF for when Byzantine workers send a gradient based on the fall of empires attack with epsilon=0.1 [Xie et. al 2020] and when they send 10x amplified sign-flipped gradients [Zue et. al 2021]. We are happy to include these in the supplement.
>
> [Xie et. al 2020] Fall of Empires: Breaking Byzantine-tolerant SGD by Inner Product Manipulation, UAI 2020.
>
> [Zue et. al 2021] Byzantine-Resilient Non-Convex Stochastic Gradient Descent, ICLR 2021.

---

> > ### Comment · Reviewer_AykZ · 2023-08-12
> >
> > Thanks for the rebuttal with a detailed explanation. The authors have addressed my concerns to some extent through the response, so I will raise the score by one level.

---

> > > ### Author Response · Authors · 2023-08-20
> > >
> > > We appreciate you taking the time to review our response and raising your score. Please let us know if you have any further questions or need clarification. We are happy to address them.

---

### Official Review · Reviewer_1ghS · 2023-07-08

**Soundness:** 1 poor
**Presentation:** 1 poor
**Contribution:** 2 fair
**Rating:** 3
**Confidence:** 4

**Summary:**

This paper tackles the problem of Byzantine robustness in distributed learning by proposing a new robust aggregation rule called Flag Aggregator. The latter is based on maximum likelihood estimation with regularization. They empirically show that using distributed gradient descent with Flag Aggregator performs well against simulated Byzantine attacks compared to other existing solutions.

**Strengths:**

The problem of Byzantine robustness is important in distributed learning. Moreover, the proposed Flag Aggregator seems to follow a creative approach.

**Weaknesses:**

My main concern is the great lack of clarity of the paper, especially in the theoretical part. I also think that the theoretical and experimental parts lack several elements.

* Lack of clarity: the paper has several clarity-affecting issues which makes it really hard to assess the technical contributions.
	* The paper starts (right away) with an unclear optimization problem (Equation 1): what are A, Y and C?
	* line 99: why is $Y Y^\top G$ a "reconstruction" of $G$? and what is meant by reconstruction exactly?
	* lines 100-103: I could not verify the stated claims/intuitions.
	* line 116: why does orthogonality imply efficiency? Authors seem to say that it is because we can derive a one-rank matrix factorization, but this does not require orthogonality of the matrix. In fact, $YY^\top G$ is just $G$ if $Y$ is orthogonal.
	* lines 123-135: this paragraph assumes that the reader knows what the Flag/Grassmanian manifold is, which was not the case for me.
	* Section 2.2: where does the vector $v$ come from? It is directly sent by the workers? Also, why do you assume that it follows a Beta distribution?
	* Algorithm 1: I could not find IRLS explained in the text. Also, it is strange that workers locally perform the update step. It always happens at server level in distributed SGD.
	* line 163: what is Flag Median?
	* line 188: what is a "second order optimal local solution"?

* Lack of convergence guarantees: After all, a Byzantine-robust learning solution should have convergence guarantees, since simulated attacks are not guaranteed to be optimal; i.e. instantiate worst-case adversaries. Typically [Karimireddy et al. 2022, Allouah et al. 2023], convergence to a neighborhood of the original solution is ensured in the presence of Byzantine workers for smooth non-convex losses.

* Experimental section: I suggest simulating more Byzantine attacks. The tested attacks (uniformly random vectors) are extremely weak compared to FoE [Xie et al. 2020], ALIE [Baruch et al. 2019] and others, which is unfortunate since the paper consider Byzantine adversaries. Also, some advanced defenses like NNM [Allouah et al. 2023] and Bucketing [Karimireddy et al. 2022] are missing although they were intended for non-iid; it is important to check how they perform against your method to assess the significance of the contribution.

[Allouah et al. 2023] Fixing by Mixing: A Recipe for Optimal Byzantine ML under Heterogeneity, AISTATS 2023.

[Karimireddy et al. 2022] Byzantine-Robust Learning on Heterogeneous Datasets via Bucketing, ICLR 2022.

[Xie et al. 2020] Fall of Empires: Breaking Byzantine-tolerant SGD by Inner Product Manipulation, UAI 2020.

[Baruch et al. 2019] A Little Is Enough: Circumventing Defenses For Distributed Learning, NeurIPS 2019.

**Questions:**

I suggest that the authors address the weaknesses listed above.

**Limitations:**

Yes.

---

> ### Author Rebuttal · Authors · 2023-08-09
>
> **Q.** In Equation 1, what are $A$, $Y$ and $C$?
>
> **Ans.** $A$ denotes the aggregation function, $Y$ denotes the decision variable in the optimization problem, and $C$ denotes the desired constraints. The reviewer will note that later in the paper, we have explicitly defined what these are in equation (5) -- where $A$ is the sum of the loglikelihood and regularization terms and $C$ is the set of matrices with orthonormal columns. We are happy to clarify this near equation (5).
>
> **Q.** Regarding reconstruction at line 99 and orthogonality's efficiency implication at line 116.
>
> **Ans.** Given gradient matrix $G$ and subspace $Y$, projection of $G$ onto $Y$ is given by $\mathbf{P}=Y^TG$. The entry $\mathbf{P}_{ji}$ has the amount (measured using dot product) of $g_i$ along $y_j$. So, $YP$ gives us the reconstruction of $G$ using each column of $Y$. This two step corresponds to reconstruction of $G$ using $Y$.
>
> Formal proof: This is folklore result that can be found in various places, but we provide a formal proof here for completeness sake. By reconstruction, we mean that the matrix $YY^TG$ is the best (or optimal) $m-$rank reconstruction of $G$ -- here optimality is with respect to Squared $\ell_2$ norm which is also known as Mean Reconstruction Error (MSE). In detail, we are given with gradient matrix $G$, and  $y_j,j=1,...,m$ such that  $y_j$'s are orthonormal, that is, $y_j^Ty_{j'}=1$ if $j=j'$, and $0$ otherwise. Since each column of $G$ is multiplied by the aggregation matrix $YY^T$ separately, we consider each $g_i$ individually.
>
> (i) Case 1: \(m=1\), so we are given with just one \(y\) such that $\left\|| y \right\||_2=1$. Then projecting $g_i$ onto $y$ in MSE is the solution to a 1-d optimization problem:
>
> \begin{equation}
> \arg\min_{\mathbf{p}\in\mathbb{R}}\left[MSE(\mathbf{p}):=\left\||g_i-\mathbf{p} y\right\||_2^2 =\left\||g_i\right\||^2-2 \mathbf{p} g_i^{T} y+\mathbf{p}^2\left\||y\right\||_2^2\right] =\frac{g_i^{T} y}{\left\||y\right\||_2^2}=g_i^{T} y,
> \end{equation}
>
> where we used the fact that $\||y\||_2=1$ in the last line. So the reconstruction is given by scaling $y$ by the optimal $\mathbf{p}=g_i^{T} y$.
> It turns out that this calculation can be performed with each basis as we show in the next case.
>
> (ii) Case 2: $m>1$, so we are given $m$ pairwise orthonormal vectors and similar to previous case we have to determine the $m$ projection coefficients for each $g_i$.
> Given $g_i$, we determine $\mathbf{p}\in\mathbb{R}^m$ as follows:
>
> \begin{equation}
> \arg\min_{\mathbf{p}_1, \cdots, \mathbf{p}_m}\left[MSE(\mathbf{p}_1,\cdots,\mathbf{p}_m):=\left\|\left|g_i - \sum\_{j=1}^m \mathbf{p}_j y_j\right\|\right|_2^2=\left\||g_i\right\||_2^2-2 \sum\_{j=1}^m \mathbf{p}_j g_i^{T} y_j+\sum\_{j=1}^m \mathbf{p}_j^2\left\||y_j\right\||_2^2\right]
> \end{equation}
>
> where we used orthogonality relationship in the last equality.
> By setting $\nabla_{\mathbf{p}_j}(MSE)=0$ we see that the reconstruction problem decomposes to $m$ 1-d optimization problems each with closed form solutions $\mathbf{p}_j=\frac{g_i^{T} y_j}{\left\|y_j\right\|^2}=g_i^{T} y_j, j=1,\dots,m$ as in the previous case. So in this case, the reconstruction is given by $\sum_j{\mathbf{p}_jy_j}=\sum_jy_jy_j^Tg_i =Y Y^{T} g_i$
> We hope this illustrates why we require orthogonality constraints since otherwise, reconstruction might be computationally expensive. Note that $Y^{T} Y=I$ does not imply $Y Y^{T}=I$
> since $m<n$. In literature, the matrix $YY^T$ is often called the family of Projection matrices (not the $Y^TG$ as we do here) since $(YY^T)^2=YY^TYY^T=YIY^T=YY^T$ for any orthonormal $Y$.
>
> **Q.** Lines 123-135: Regarding Flag/Grassmanian manifold.
>
> **Ans.** We will add the necessary background in the appendix.
>
> **Q.** Section 2.2: Regarding v and if we assume that it follows a Beta distribution.
>
> **Ans.** As indicated, $v\in[0,1]$ is a value in between $0$ and $1$.
> Beta distribution is usually used in economics to model robustness which is one reason we chose it.  Beta distribution can be used to model distributions with various types of information involving skewness which we believe is convenient since the priors are easy to set, which sometimes can be crucial for aggregation purposes.
>
> **Q.** Algorithm 1: IRLS explanation and where the update step is performed in distributed SGD.
>
> **Ans.** Please see the general response for a description of IRLS and its connection to Flag Aggregator. The model updates are done locally by the workers after they receive the aggregated update from the server.
>
> **Q.** Line 163: what is Flag Median?
>
> **Ans.** It is defined as a specific type median of subspaces as proposed in [Mankovich et al. 2022].
>
> **Q.** Line 188: what is a ``second order optimal local solution''?
>
> **Ans.** Feasible points such that the Hessian has a nonnegative curvature -- all eigenvalues are nonnegative.
>
> **Q.** A Byzantine-robust learning solution should have convergence guarantees since simulated attacks are not guaranteed to be optimal; i.e. instantiate worst-case adversaries.
>
> **Ans.** Please note that the matrix $YY^T$ is a symmetric positive semidefinite matrix, so our method is guaranteed to converge whenever the original architecture algorithm converges since such matrices have eigendecomposition with all nonnegative eigenvalues. Intuitively, applying $YY^T$ on $G$ simply corresponds to scaling different parameter gradients after rotation with the eigenvectors of $YY^T$, similar to a precondition.
>
> **Q.** Regarding the experimental section.
>
> **Ans.** These papers are creative and very interesting. We will consider them in future work but there is a crucial difference: they do **not** formulate their aggregation scheme as an optimization problem that immediately can be transformed into a computational problem, as we have done. Moreover, these methods are often analyzed using sophisticated assumptions whereas convergence of our method can be guaranteed under standard assumptions in optimization literature.

---

> > ### Comment · Reviewer_1ghS · 2023-08-13
> >
> > I have read the authors' rebuttal and other reviews, and I am maintaining my score.

---

> > > ### Author Response · Authors · 2023-08-20
> > >
> > > We are grateful for your time in reviewing our response and other feedback. If you have any further questions or require clarification, please do not hesitate to inform us. We are happy to provide the answers you need.

---

### Official Review · Reviewer_Pt2v · 2023-07-08

**Soundness:** 3 good
**Presentation:** 3 good
**Contribution:** 3 good
**Rating:** 6
**Confidence:** 3

**Summary:**

Authors propose a gradient aggregation method for distributed optimization that is robust to Byzantine device failures in large scale distributed setups. In each round, given the set of gradients from each workers, the authors aim to find the optimal low-rank subspace that can explain the variance of a majority of the gradients. The authors formulate the problem as a MLE under a beta distribution setup  and solve an approximate version of the problem though SDP.

**Strengths:**

Byzantine device failures is an important concern for large scale clusters.  The presented method is well motivated theoretically and backed up with experiments comparing their robustness properties to other aggregation methods. Results demonstrate a significant advantage of this aggregation setup.

**Weaknesses:**

Although it is evident that Byzantine failures can have a significant impact on gradient computation if using simple aggregation rules, its unclear how often such failures happen in the cluster sizes the authors have considered. Augmentation pipelines induce their own noise to gradient information, but its unclear if these will be adversarial in _each_ update step. The amount of noise induced and its effect on adversarial training setups is also not evident.  (See questions). This makes it unclear how the clear advantages of the method translates to real-world workloads especially considering that the method adds a potentially expensive top-k SVD computation step.

**Questions:**

- Frequency of Byzantine failures: Could you provide some insights into how frequent failures due to hardware/software/augmentation pipeline based issues occur in training runs.  Assuming there will be at least a single byzantine worker at all times (i.e $f\ge1$ in Fig 4) seems too strong and can be better contextualized with some supporting evidence.
- Scalability of method to federated clusters: Byzantine failures will potentially be a larger concern when training over heterogeneous hardware and partially available clients, for ex. in the federated clusters. Could the authors comment on the feasibility of running the method in such settings, considering that the majority of the computation is performed at the central server?

**Limitations:**

Yes

---

> ### Author Rebuttal · Authors · 2023-08-09
>
> **Q.** Could you provide some insights into how frequent failures due to hardware/software/augmentation pipeline based issues occur in training runs.
>
> **Ans.** Training today’s large models is a very time-consuming task that can take days or even weeks. An important problem is the fact that failures inside the training environment, e.g. a datacenter, impede the progress of distributed training. An analysis from Microsoft that spans across two months and uses around 100k jobs run by hundreds of users was presented in [Jeon et. al 2019]. As a high-level summary, jobs using more than 4 GPUs, finish unsuccessfully at higher rate due to various reasons (Section 4.2). When the jobs fail, they waste a lot of computing time which is also analyzed in another study from Facebook [Eisenman et. al 2022] on a system comprising 21 clusters over a period of one month. It is important to note that these training jobs interact with multiple systems during the training process, such as accessing training samples from a separate reader cluster. Consequently, any failure within these interconnected systems will impede the overall progress of the training.
>
> **Q.** Could the authors comment on the feasibility of running the method in federated clusters setting, considering that the majority of the computation is performed at the central server?
>
> **Ans.** In Fig. 9 we tested the scalability of FA to a larger cluster within our hardware resources. For federated clusters, our one-cluster setup could be extended in a hierarchical architecture where we would have gradient-computing workers sending their results to the representative aggregating workers (which play the role of a PS for that cluster). Aggregating workers would further combine the partially aggregated results with other clusters representatives in another level of hierarchy. This allows scaling FA to federated clusters, however, the implementation of this approach is beyond the scope of our paper.
>
> [Jeon et. al 2019] Analysis of Large-Scale Multi-Tenant GPU Clusters for DNN Training Workloads, ATC 2019.
>
> [Eisenman et. al 2022] Check-N-Run: a Checkpointing System for Training Deep Learning Recommendation Models, NSDI 2022.

---

> > ### Comment · Reviewer_Pt2v · 2023-08-18
> > **Re**
> >
> > Thank you for your comment. I maintain my positive assessment of the work.

---

> > > ### Author Response · Authors · 2023-08-20
> > >
> > > Thank you for taking the time to review our response. We appreciate your positive assessment of our work. We are more than happy to answer any further questions or clarification you might have.

---

### Official Review · Reviewer_rLy5 · 2023-07-16

**Soundness:** 2 fair
**Presentation:** 2 fair
**Contribution:** 2 fair
**Rating:** 6
**Confidence:** 3

**Summary:**

The paper proposes a new appraoch for aggregating gradients for distributed ML training under Byzantine failures, noise due to data augmentation, etc. The approach relies on constructing a low-dimensional subspace such that the proportion of variance of the gradient vectors contained in the subspace is maximized. The authors derive the loss function for their setting and formulate the problem as a regularized convex optimization problem which can be solved with standard solvers to obtain the basis for the subspace. The update direction is then obtained by projecting the individual gradients onto the basis and then averaging the result. Experiments on different datasets and number of workers show improved prediction accuracy over baselines when distributed training is performed using the proposed method.

**Strengths:**

1. The proposed approach is principled and easy to interpret as it tries to identify the subspace which contains the maximum proportion of the variance of the gradients and is also easy to implement due to its formulation as a regularized convex optimization problem which can be solved by off-the-shelf workers.

2. The approach is extensively evaluated on a range of datasets (MNIST, CIFAR10, tiny-Imagenet) and for different noise models (random noise, adversarial data augmentation etc). I also appreciate the authors presenting results on wall-clock time to accuracy and per-iteration time thereby acknowledging the extra time required per iteration in their approach to compute the aggregated gradients. This opens the door to future research on speeding up the proposed aggregation method while retaining the accuracy gains.

**Weaknesses:**

1. My main concern with the approach is its novelty. Since the goal appears to be to estimate the subspace containing the maximum proportion of gradient variance, I am not sure why this cannot be done by retaining the top-k Principal Components of the gradients. The authors even acknowledge in line 109 that the idea to use the ratio of variance of projected and true gradients has been explored in the Robust PCA literature. However, they do not explain why simply considering the principal components will not work, nor do they perform experiments with PCA/Robust PCA as baselines. I would like to see at least one of the two (explanation/experiments) to be convinced of the need for the proposed approach and its gains over PCA.

2. The extra computational cost and the added time per iteration as seen in Fig 10 (b) is also a weakness. While I do appreciate the authors measuring and presenting this time, it is not clear at this point if the accuracy gain justifies the extra time. One way to demonstrate this would be to allow the other approaches to run for the same amount of time in Fig 10 (c). If it could be shown that even after running for that long these approaches cannot match the accuracy of Flag Aggregation, then the extra time required could be justified.

**Questions:**

1. Could you please explain more clearly (preferably with an example) why adversarial training could lead to noise in gradients? The current explanation in lines 59-61 is too vague and high-level. Clarifying this would be useful for readers not familiar with the adversarial training literature, and would strengthen the motivation of the approach.

2. Please introduce/explain the term Flag Optimization before using it in line 75, or add a citation since I don't think readers outside the optimization community would be familiar with this term.

3. Fig. 5 seems to suggest that Flag Aggregation is only useful for bs >= 128? Is that indeed the case? What is the value of bs in the other experiments?

4. Likewise Fig. 6 seems suggest that it is useful only for p >= 11. Please clarify if that is indeed the case. Note that, gains only in certain regimes of bs and p won't necessarily be a reason for rejection. But it is important to acknowledge it in the paper so that the readers have all the information.

5. In line 119 you mention that gradient quality from workers may differ if workers use different batch size. I think this is a very interesting and practical scenario. Did you perform any experiments where workers used different batch sizes? Will it be possible to present the behaviour of Flag Aggregation and other baselines in this scenario?

6. Can methods from randomized linear algebra, or other approaches to speed up SVD help in reducing the per-iteration time of your approach? If yes, it might be worth mentioning this in the paper as an option for readers looking to implement your approach.

**Limitations:**

I feel the main limitations of the work are the increased computation time per iteration and the lack of clarity on novelty w.r.t PCA. I appreciate the authors' acknowledgement of the higher per-iteration time and look forward to their responses to the other limitations that I have identified under Weaknesses, above.

---

> ### Author Rebuttal · Authors · 2023-08-09
>
> **Q.** Why simply considering the principal components will not work? Did we perform experiments with PCA/Robust PCA as baselines?
>
> **Ans.** As explained in the general response, mathematically, one iteration of FA with uniform weights assigned across all workers is equivalent to PCA. The main novelty in our FA approach is the extension of PCA to an iteratively *reweighted* form that is guaranteed to converge. Specifically, we show that we obtain a convergent procedure in which we repeatedly solve weighted PCA problems. Moreover, the convergence guarantee immediately follows when the procedure is viewed as an IRLS procedure solving the MLE problem induced by the value of workers modeled with a beta distribution as in Sec 2.2. We added a baseline for top-m principal components of the gradient matrix in Fig 1(b) of the attached pdf to the global response.
>
> **Q.** It is not clear at this point if the accuracy gain justifies the extra time. One way to demonstrate this would be to allow the other approaches to run for the same amount of time in Fig 10 (c). If it could be shown that even after running for that long these approaches cannot match the accuracy of Flag Aggregation, then the extra time required could be justified.
>
> **Ans.** Thank you for your suggestion. Although FA gains are becoming visible towards the end of Fig 10(a) which is the zoomed-in version of Fig 10(c), we’re happy to let the other approaches run equally as long in terms of wall clock time, and we show the consistency of our results for this longer timespan in Fig 1(a) inside the attached PDF. We can also include this figure in the final version of the paper if needed.
>
> **Q.** Why adversarial training could lead to noise in gradients?
>
> **Ans.** Intuitively, the goal of adversarial training seems to be to make models predict all the nearby samples accurately, given the training set. The so-called Adversarial samples are typically constructed by introducing small imperceptible perturbations to the original data that would lead the model to make incorrect predictions. These perturbations are calculated based on gradients obtained from the model itself with respect to training set samples. Due to the complexity of the models and the non-linear nature of the deep network functions, or as training proceeds, it gets more challenging to find such adversarial samples. Recent technical results indicate that there are randomized algorithms that provide adversarial robust guarantees in expectation *only*.
> Hence, these randomized algorithms, by design, have a failure probability. Our method could be used when the knowledge of how these adversarial sample-generating frameworks behave, is not fully understood and the generated models are difficult to understand. We are very keen on exploring this aspect in our future work!
>
> **Q.** Introduce/explain the term Flag Optimization before using it in line 75, or add a citation.
>
> **Ans.** Thank you for pointing this out. We will add a citation to clarify this before explaining more on line 123.
>
> **Q.** Fig. 5 seems to suggest that Flag Aggregation is only useful for $bs \geq 128$? Is that indeed the case? What is the value of bs in the other experiments?
>
> **Ans.** We have an experiment that discusses the utility of larger batch sizes at line 268. As mentioned on line 236, the batch size across experiments is fixed to $128$ unless otherwise stated.
>
> **Q.** Fig. 6 seems to suggest that it is useful only for $p \geq 11$. Please clarify if that is indeed the case.
>
> **Ans.** Our FA framework does not require that $p \geq 11$ and is not a specific choice for our design. As mentioned in Section 3.1 (Testbed), from a technical perspective related to our hardware resources, we instantiate $p=15$ workers unless otherwise stated. Our intention of having smaller $p$ values in the experiment related to Fig. 6 was to evaluate the marginal utility of having more workers under a fixed amount of noise $(f=2)$. We could not test for $p \leq 11$ using other baselines such as Bulyan (which requires $p \geq 4f+3$ as mentioned on line 193 for its best performance), so we decided to leave out those experiments. We will clarify this in the experiment.
>
> **Q.** Did you perform any experiments where workers used different batch sizes? Will it be possible to present the behavior of Flag Aggregation and other baselines in this scenario?
>
> **Ans.** Thank you for your suggestion. In our experiments, batch sizes are fixed across workers. However, our current framework allows using different batch sizes in workers at line 3 using: (i) the average of local gradients at worker $i$, and/or (ii) directly adding them in the SVD computation. Our experiments are with local averaging since we have limited GPU availability at our disposal.
>
> **Q.** Can methods from randomized linear algebra, or other approaches to speed up SVD help in reducing the per-iteration time of your approach?
>
> **Ans.** Yes, thank you for your suggestion. We will clarify this more in section 4 of the paper. Please refer to our general answer for more detail.

---

> > ### Comment · Reviewer_rLy5 · 2023-08-17
> > **Re**
> >
> > Thank you for the response. I am satisfied with the responses to all my questions, and I also appreciate the effort put into the additional experiments performed to substantiate your claims in response to the points I had identified under weaknesses. I would definitely recommend including these plots in the final version of the paper or appendix, if accepted. I am increasing my score to 6 (Weak Accept). I do not have any other questions or concerns.

---

> > > ### Author Response · Authors · 2023-08-20
> > >
> > > Thank you for taking the time to review our response and for increasing your score. We are pleased that our responses met your expectations, and we will definitely incorporate the recommended plots in the camera-ready version.

---

### Official Review · Reviewer_Zke2 · 2023-07-25

**Soundness:** 3 good
**Presentation:** 2 fair
**Contribution:** 3 good
**Rating:** 6
**Confidence:** 3

**Summary:**

This paper presents a new method to aggregate gradients in a distributed training setting. Effectively, the proposed algorithm projects gradients onto a learned lower dimensional subspace and then aggregates the projections using standard techniques like averaging. This leads to a more robust aggregation against Byzantine failures. The projection is similar to a robust PCA, and is learnt using an approximate MLE via a Taylor expansion, leading to a computationally more feasible algorithm. Thorough experiments are conducted that demonstrate the efficacy of the proposed algorithm.

**Strengths:**

1. The proposed novel algorithm empirically performs better than existing methods when measured by iteration complexity.
2. The authors provide a thorough comparison to existing methods and place their work in context.

**Weaknesses:**

1. The exposition in the paper lacks clarity in some places -- for example, the IRLS subroutine in Algorithm 1 is not described or even briefly summarized in the main paper.
2. The authors do not present their theoretical convergence results in the main body of the paper.
3. As pointed out by the authors, the main proposed algorithm does not seem to perform significantly better than other existing algorithms when comparing wall clock runtimes.

**Questions:**

1. What exactly is the IRLS procedure in Algorithm 1? There is no description in the main body of this subroutine or procedure.
2. In algorithm 1, line 5, the procedure to find the approximate subspace $\hat Y$ is done only on the server. Presumably, this procedure would be the bottleneck in the whole algorithm, since it involves multiple SVD computations. Can the authors comment on the breakdown of which parts of algorithm 1 take significantly more time, and explain any optimizations they have implemented in this context?

**Limitations:**

Yes, the authors addressed the limitations of their work in the paper.

---

> ### Author Rebuttal · Authors · 2023-08-09
>
> **Q.** What exactly is the IRLS procedure in Algorithm 1?
>
> **Ans.** Answered in the general response.
>
> **Q.** Can the authors comment on the breakdown of which parts of algorithm 1 take significantly more time, and explain any optimizations they have implemented in this context?
>
> **Ans.** This is a great question. When using FA for the aggregation phase of distributed SGD, the computation cycles are mostly spent on the IRLS procedure at line 5 of Algorithm 1. Specifically, calculating the SVD of $G^TD^{1/2}$ (or eigenvalues of the matrix $G^TDG$) contributes to most of these cycles. For more detail, please refer to our general response.

---

> > ### Comment · Reviewer_Zke2 · 2023-08-17
> >
> > Thank you for your response. I have gone through the rebuttals and the other reviews, and will keep my score unchanged.

---

> > > ### Author Response · Authors · 2023-08-20
> > >
> > > We appreciate your time in reviewing our response and other feedback. Please do not hesitate to let us know if you have any further questions or need clarification. We are happy to answer them.

---

### Author Rebuttal · Authors · 2023-08-09

We thank the reviewers for spending time going through our submission in great detail, very insightful comments, and also pointing to aspects in the presentation style that can be improved. We are glad that the reviewers find our subspace based aggregation algorithm to be novel, can be derived using standard maximum likelihood estimation principles, and is currently unavailable in the context of training deep learning models. Here below we answer two questions that we find in various flavors in some reviews, and then answer all individual questions below individual reviews. We also provided additional empirical results that were requested by reviewers on some more attacks and baselines in the pdf.

**Q.** Can you provide a brief summary of IRLS procedure for Flag Aggregation in the main paper?

**Ans.** Yes! IRLS is a standard optimization technique in which we substitute general norm functions with weighted euclidean norm functions. The key advantage of this substitution is that we may obtain closed form solution to the substituted euclidean norm version. Starting from a (random) feasible point $Y_{\text{old}}$, the weights are calculated with the general norm functions. Then, the solution $Y_{\text{new}}$ to the weighted euclidean norm optimization is obtained.  This corresponds to one iteration in IRLS and repeating the above step with this new $Y_{\text{new}}$ corresponds to the IRLS algorithm.

For aggregation purposes in FA, in each iteration the square root function or more generally, the $a-$th root function in equation (4) is replaced by reweighted quadratic function which has a closed-form solution given by SVD. Specifically, in FA, $Y_{\text{new}}$ is calculated by solving the lagrangian equation (14) {\bf in supplement} which is equivalent to computing the Singular Decomposition of a matrix defined using $GDG^T,\lambda \nabla\mathcal{R}(Y_{\text{old}})$. For our data-dependent regularization $\mathcal{R}$, this eigenvalue computation is equivalent to SVD of $G$ concatenated with weighted columns of $g_i-g_j$ (as was done for individual $g_i$'s). The proof of this equivalence can be seen in [Mankovich et al. 2022], for example where left singular vectors of $GD^{1/2}$ are used in the solution procedure. For mathematical norms mentioned in the main paper including elementwise norms involving $\ell_1$, we handle it columnwise wrt $Y\in\mathbb{R}^{n\times m}$ since each column of $Y$ corresponds to a basis vector in $\mathbb{R}^n$ of the $m-$dimensional subspace. By using a smooth approximation to $\ell_1$, for example as in Sec 1.2 in [Ene et al. 2019], we can see that a quadratic approximation is available $\mathcal{R}(Y) = \sum_{j=1}^my_j^TD_jy_j$ where $y_j\in\mathbb{R}^n$ is the $j$-th column on $Y$ and $D_j \in \mathbb{R}_{>0}^n$ is a diagonal matrix with positive entries along the diagonal calculated using $Y\_{\text{old}}$, and so can be handled similar to the data-dependent regularization. We will make space and include a description in the main paper.

**Q.**  To implement FA, is it possible to take advantage of fast, randomized SVD solvers? If so, how?

**Ans.** Yes, it is indeed possible to use existing solvers to solve FA for aggregation purposes. This is the main advantage of our FA algorithm. In detail, to calculate the left singular values of $GD^{1/2}\in\mathbb{R}^{n\times p}$, we use the fact that number of workers $p\ll n$ and solve the $p\times p$ eigenvalue problem which can be fast in practice. Upon receiving the right singular vectors, first order methods can be used to obtain the left singular vectors. In this sense, we can use any fast, randomized SVD algorithm to solve for the right and/or left singular vectors.


[Mankovich et al. 2022] The Flag Median and FlagIRLS, CVPR 2022.

[Ene et al. 2019] Improved Convergence for $\ell\_1 $ and $ \ell_{\infty} $ Regression via Iteratively Reweighted Least Squares

---

### Decision · Program_Chairs · 2023-09-21

**Decision:**

Reject

**Comment:**

This paper presents a method to aggregate gradients in a distributed training setting via projection onto lower dimensional substances. This yields more robust updates (e.g. under Byzantine attacks). The proposed approach is principled and rigorous, however, the claimed properties are not theoretically proven, but mostly evaluated in empirically.

One main point criticized by the reviewers is a lack of clarity in the manuscript. E.g. terms like 'flag manifold', 'IRLS', etc. might be difficult to asses for non-expert readers, and the same might also wonder about other (well-known) approaches such as PCA/SVD. While these relations have been clarified to the reviewers in the rebuttal, it seems to me that a full revision is better suited to properly address all these concerns of the reviewers.

The reviewers found the submission could be substantially strengthened by either theoretical results on (some of) the robustness properties, or empirical evaluation against stronger attacks.